# Multi-Block Color-Binarized Statistical Images for Single-Sample Face Recognition

**DOI:** 10.3390/s21030728

**Published:** 2021-01-21

**Authors:** Insaf Adjabi, Abdeldjalil Ouahabi, Amir Benzaoui, Sébastien Jacques

**Affiliations:** 1Department of Computer Science, LIMPAF, University of Bouira, Bouira 10000, Algeria; i.adjabi@univ-bouira.dz; 2Polytech Tours, Imaging and Brain, INSERM U930, University of Tours, 37200 Tours, France; 3Department of Electrical Engineering, University of Bouira, Bouira 10000, Algeria; a.benzaoui@univ-bouira.dz; 4GREMAN UMR 7347, University of Tours, CNRS, INSA Centre Val-de-Loire, 37200 Tours, France; sebastien.jacques@univ-tours.fr

**Keywords:** biometrics, face recognition, single-sample face recognition, binarized statistical image features, K-nearest neighbors

## Abstract

Single-Sample Face Recognition (SSFR) is a computer vision challenge. In this scenario, there is only one example from each individual on which to train the system, making it difficult to identify persons in unconstrained environments, mainly when dealing with changes in facial expression, posture, lighting, and occlusion. This paper discusses the relevance of an original method for SSFR, called Multi-Block Color-Binarized Statistical Image Features (MB-C-BSIF), which exploits several kinds of features, namely, local, regional, global, and textured-color characteristics. First, the MB-C-BSIF method decomposes a facial image into three channels (e.g., red, green, and blue), then it divides each channel into equal non-overlapping blocks to select the local facial characteristics that are consequently employed in the classification phase. Finally, the identity is determined by calculating the similarities among the characteristic vectors adopting a distance measurement of the K-nearest neighbors (K-NN) classifier. Extensive experiments on several subsets of the unconstrained Alex and Robert (AR) and Labeled Faces in the Wild (LFW) databases show that the MB-C-BSIF achieves superior and competitive results in unconstrained situations when compared to current state-of-the-art methods, especially when dealing with changes in facial expression, lighting, and occlusion. The average classification accuracies are 96.17% and 99% for the AR database with two specific protocols (i.e., Protocols I and II, respectively), and 38.01% for the challenging LFW database. These performances are clearly superior to those obtained by state-of-the-art methods. Furthermore, the proposed method uses algorithms based only on simple and elementary image processing operations that do not imply higher computational costs as in holistic, sparse or deep learning methods, making it ideal for real-time identification.

## 1. Introduction

Generally speaking, biometrics aims to identify or verify an individual’s identity according to some physical or behavioral characteristics [1]. Biometric practices replace conventional knowledge-based solutions, such as passwords or PINs, and possession-based strategies, such as ID cards or badges [2]. Several biometric methods have been developed to varying degrees and are being implemented and used in numerous commercial applications [3].

Fingerprints are the biometric features most commonly used to identify criminals [4]. The first automated fingerprint authentication device was commercialized in the early 1960s. Multiple studies have shown that the iris of the eye is the most accurate modality since its texture remains stable throughout a person’s life [5]. However, those techniques have the significant drawback of being invasive, which significantly restricts their applications. Besides, iris recognition remains problematic for users who do not wish to put their eyes in front of a sensor. On the contrary, biometric recognition based on facial analysis does not pose any such user constraints. In contrast to other biometric modalities, face recognition is a modality that can be employed without any user–sensor co-operation and can be applied discreetly in surveillance applications. Face recognition has many advantages: the sensor device (i.e., the camera) is simple to mount; it is not costly; it does not require subject co-operation; there are no hygiene issues; and, being passive, people much prefer this modality [6].

Two-dimensional face recognition with Single-Sample Face Recognition (SSFR) (i.e., using a Single- Sample Per Person (SSPP) in the training set) has already matured as a technology. Although the latest studies on the Face Recognition Grand Challenge (FRGC) [7] project have shown that computer vision systems [8] offer better performance than human visual systems in controlled conditions [9], research into face recognition, however, needs to be geared towards more realistic uncontrolled conditions. In an uncontrolled scenario, human visual systems are more robust when dealing with the numerous possibilities that can impact the recognition process [10], such as variations in lighting, facial orientation, facial expression, and facial appearance due to the presence of sunglasses, a scarf, a beard, or makeup. Solving these challenges will make 2D face recognition techniques a much more important technology for identification or identity verification.

Several methods and algorithms have been suggested in the face recognition literature. They can be subdivided into four fundamental approaches depending on the method used for feature extraction and classification: holistic, local, hybrid, and deep learning approaches [11]. The deep learning class [12], which applies consecutive layers of information processing arranged hierarchically for representation, learning, and classification, has dramatically increased state-of-the-art performance, especially with unconstrained large-scale databases, and encouraged real-world applications [13,14].

Most current methods in the literature use several facial images (samples) per person in the training set. Nevertheless, in real-world systems (e.g., in fugitive tracking, identity cards, immigration management, or passports), only SSFR systems are used (due to the limited storage and privacy policy), which employ a single sample per person in the training stage (generally neutral images acquired in controlled conditions), i.e., just one example of the person to be recognized is recorded in the database and accessible for the recognition task [15]. Since there are insufficient data (i.e., we do not have several samples per person) to perform supervised learning, many well-known algorithms may not work particularly well. For instance, Deep Neural Networks (DNNs) [13] can be used in powerful face recognition techniques. Nonetheless, they necessitate a considerable volume of training data to work well. Vapnik and Chervonenkis [16] showed that vast training data must ensure learning systems’ generalization in their statistical learning theorem. In addition, the use of three-dimensional (3D) imaging instead of two-dimensional representation (2D) has made it possible to cover several issues related to image acquisition conditions, in particular pose, lighting and make-up variations. While 3D models offer a better representation of the face shape for a clear distinction between persons [17,18], they are often not suitable for real-time applications because they require expensive and sophisticated calculations and specific sensors. We infer that SSFR remains an unsolved issue in academic and business circles, particularly with respect to the major efforts and growth in face recognition.

In this paper, we tackle the SSFR issue in unconstrained conditions by proposing an efficient method based on a variant of the local texture operator Binarized Statistical Image Features (BSIF) [19] called Multi-Block Color-binarized Statistical Image Features (MB-C-BSIF). It employs local color texture information to obtain honest and precise representation. The BSIF descriptor has been widely used in texture analysis [20,21] and has proven its utility in many computer vision tasks. In the first step, the proposed method uses preprocessing to enhance the quality of facial photographs and remove noise [22,23,24]. The color image is then decomposed into three channels (e.g., red, green, and blue for the RGB color-space). Next, to find the optimum configuration, several multi-block decompositions are checked and examined under various color-spaces (i.e., we tested RGB, Hue Saturation Value (HSV), in addition to the YCbCr color-spaces, where Y is the luma component; Cb and Cr are the blue-difference and red-difference chroma components, respectively). Finally, classification is undertaken using the distance measurement of the K-nearest neighbors (K-NN) classifier. Compared to several related works, the advantage of our method lies in exploiting several kinds of information: local, regional, global, and color-texture. Besides, the algorithm of our method is simple and does not require greater complexity, which makes it suitable for real-time applications (e.g., surveillance systems or real-time identification). Our system is based on only basic and simple image processing operations (e.g., median filtering, a simple convolution, or histogram calculation), involving a much lower computational cost than existing systems. For example, (1) Subspace or sparse representation-based methods involve many calculations and higher time in dimensionality reduction, or (2) Deep learning methods involve very high complexity cost and require many computations. For such systems, GPUs’ need clearly shows that many calculations must be done in parallel; GPUs are designed to run concurrently with thousands of processor cores, making for extensive parallelism where each core is concentrated on making accurate calculations. With a standard CPU, a considerable amount of time for training and testing will be needed for deep learning systems.

The rest of the paper is structured as follows. We discuss relevant research about SSFR in Section 2. Section 3 describes our suggested method. In Section 4, the experimental study, key findings, and comparisons are performed and presented to show our method’s superiority. Section 5 of the paper presents key findings and discusses research perspectives.

## 2. Related Work 

Current methods designed to resolve the SSFR issue can be categorized into four fundamental classes [25], namely: virtual sample generating, generic learning, image partitioning, and deep learning methods.

### 2.1. Virtual Sample Generating Methods

The methods in this category produce some additional virtual training samples for each individual to augment the gallery (i.e., data augmentation), so that discriminative sub-space learning can be employed to extract features. For example, Vetter (1998) [26] proposed a robust SSFR algorithm by generating 3D facial models through the recovery of high-fidelity reflectance and geometry. Zhang et al. (2005) [27] and Gao et al. (2008) [28] developed two techniques to tackle the issue of SSFR based on the singular value decomposition (SVD). Hu et al. (2015) [29] suggested a different SSFR system based on the lower-upper (LU) algorithm. In their approach, each single subject was decomposed and transposed employing the LU procedure and each raw image was rearranged according to its energy. Dong et al. (2018) [30] proposed an effective method for the completion of SSFR tasks called K-Nearest Neighbors virtual image set-based Multi-manifold Discriminant Learning (KNNMMDL). They also suggested an algorithm named K-Nearest Neighbor-based Virtual Sample Generating (KNNVSG) to augment the information of intra-class variation in the training samples. They also proposed the Image Set-based Multi-manifold Discriminant Learning algorithm (ISMMDL) to exploit intra-class variation information. While these methods can somewhat alleviate the SSFR problem, their main disadvantage lies in the strong correlation between the virtual images, which cannot be regarded as independent examples for the selection of features.

### 2.2. Generic Learning Methods

The methods in this category first extract discriminant characteristics from a supplementary generic training set that includes several examples per individual and then use those characteristics for SSFR tasks. Deng et al. (2012) [31] developed the Extended Sparse Representation Classifier (ESRC) technique in which the intra-class variant dictionary is created from generic persons not incorporated in the gallery set to increase the efficiency of the identification process. In a method called Sparse Variation Dictionary Learning (SVDL), Yang et al. (2013) [32] trained a sparse variation dictionary by considering the relation between the training set and the outside generic set, disregarding the distinctive features of various organs of the human face. Zhu et al. (2014) [33] suggested a system for SSFR based on Local Generic Representation (LGR), which leverages the benefits of both image partitioning and generic learning and takes into account the fact that the intra-class face variation can be spread among various subjects.

### 2.3. Image Partitioning Methods

The methods in this category divide each person’s images into local blocks, extract the discriminant characteristics, and, finally, perform classifications based on the selected discriminant characteristics. Zhu et al. (2012) [34] developed a Patch-based CRC (PCRC) algorithm that applies the original method proposed by Zhang et al. (2011) [35], named Collaborative Representation-based Classification (CRC), to each block. Lu et al. (2012) [36] suggested a technique called Discriminant Multi-manifold Analysis (DMMA) that divides any registered image into multiple non-overlapping blocks and then learns several feature spaces to optimize the various margins of different individuals. Zhang et al. (2018) [37] developed local histogram-based face image operators. They decomposed each image into different non-overlapping blocks. Next, they tried to derive a matrix to project the blocks into an optimal subspace to maximize the different margins of different individuals. Each column was then redesigned to an image filter to treat facial images and the filter responses were binarized using a fixed threshold. Gu et al. (2018) [38] proposed a method called Local Robust Sparse Representation (LRSR). The main idea of this technique is to merge a local sparse representation model with a block-based generic variation dictionary learning model to determine the possible facial intra-class variations of the test images. Zhang et al. (2020) [39] introduced a novel Nearest Neighbor Classifier (NNC) distance measurement to resolve SSFR problems. The suggested technique, entitled Dissimilarity-based Nearest Neighbor Classifier (DNNC), divides all images into equal non-overlapping blocks and produces an organized image block-set. The dissimilarities among the given query image block-set and the training image block-sets are calculated and considered by the NNC distance metric.

### 2.4. Deep Learning Methods

The methods in this category employ consecutive hidden layers of information-processing arranged hierarchically for representation, learning, and classification. They can automatically determine complex non-linear data structures [40]. Zeng et al. (2017) [41] proposed a method that uses Deep Convolutional Neural Networks (DCNNs). Firstly, they propose using an expanding sample technique to augment the training sample set, and then a trained DCNN model is implemented and fine-tuned by those expanding samples to be used in the classification process. Ding et al. (2017) [42] developed a deep learning technique centered on a Kernel Principal Component Analysis Network (KPCANet) and a novel weighted voting technique. First, the aligned facial image is segmented into multiple non-overlapping blocks to create the training set. Then, a KPCANet is employed to get filters and banks of features. Lastly, recognition of the unlabeled probe is achieved by applying the weighted voting form. Zhang and Peng (2018) [43] introduced a different method to generate intra-class variances using a deep auto-encoder. They then used these intra-class variations to expand the new examples. First, a generalized deep auto-encoder is used to train facial images in the gallery. Second, a Class-specific Deep Auto-encoder (CDA) is fine-tuned with a single example. Finally, the corresponding CDA is employed to expand the new samples. Du and Da (2020) [44] proposed a method entitled Block Dictionary Learning (BDL) that fuses Sparse Representation (SR) with CNNs. SR is implemented to augment CNN efficiency by improving the inter-class feature variations and creating a global-to-local dictionary learning process to increase the method’s robustness.

It is clear that the deep learning approach for face recognition has gained particular attention in recent years, but it suffers considerably with SSFR systems as they still require a significant amount of information in the training set.

Motivated by the successes of the third approach, “image partitioning”, and the reliability of the local texture descriptor BSIF, in this paper, we propose an image partitioning method to address the problems of SSFR. The proposed method, called MB-C-BSIF, decomposes each image into several color channels, divides each color component into various equal non-overlapping blocks, and applies the BSIF descriptor to each block-component to extract the discriminative features. In the following section, the framework of the MB-C-BSIF is explained in detail.

## 3. Proposed Method

This section details the MB-C-BSIF method (see Figure 1) proposed in this article to solve the SSFR problem. MB-C-BSIF is an approach based on image partitioning and consists of three key steps: image pre-processing, feature extraction based on MB-C-BSIF, and classification. In the following subsections, we present these three phases in detail.

### 3.1. Preprocessing

The suggested feature extraction and classification rules compose the essential steps in our proposed SSFR. However, before driving these two steps, pre-processing is necessary to improve the visual quality of the captured image. The facial image is enhanced by applying histogram normalization and then filtered with a non-linear filter. The median filter [45] was adopted to minimize noise while preserving the facial appearance and enhancing the operational outcomes [46].

### 3.2. MB-C-BSIF-Based Feature Extraction

Our advanced feature extraction technique is based on the multi-block color representation of the BSIF descriptor, entitled Multi-Block Color BSIF (MB-C-BSIF). The BSIF operator proposed by Kannala and Rahtu [16] is an efficient and robust descriptor for texture analysis [47,48]. BSIF focuses on creating local image descriptors that powerfully encode texture information and are appropriate for describing image regions in the form of histograms. The method calculates a binary code for all pixels by linearly projecting local image blocks onto a subspace whose basis vectors are learned from natural pictures through Independent Component Analysis (ICA) [45] and by binarizing the coordinates through thresholding. The number of basis vectors defines the length of the binary code string. Image regions can be conveniently represented with histograms of the pixels’ binary codes. Other descriptors that generate binary codes, such as the Local Binary Pattern (LBP) [49] and the Local Phase Quantization (LPQ) [50], have inspired the BSIF process. However, the BSIF is based on natural image statistics rather than heuristic or handcrafted code constructions, enhancing its modeling capabilities.

Technically speaking, the $s_{i}$ filter response is calculated, for a given picture patch X of size l×l pixels and a linear filter $W_{i}$ of the same size, by:(1)$$s_{i}=\underset{u,v}{\sum }W_{i}\left(u,v\right)X\left(u,v\right)\;$$
where the index i in $W_{i}$ indicates the $i^{th}$ filter.

The binarized $b_{i}$ feature is calculated as follows:(2)$$b_{i}=\left\{\begin{array}{c} 1\;\;\;\;\mathrm{if}\;s_{i}>0\;\\ 0\;\mathrm{otherwise} \end{array}\right.$$

The BSIF descriptor has two key parameters: the filter size l×l and the bit string length  n. Using ICA, $W_{i}$ filters are trained by optimizing $\;s_{i}$’s statistical independence. The training of $W_{i}$ filters is based on different choices of parameter values. In particular, each filter set was trained using 50,000 image patches. Figure 2 displays some examples of the filters obtained with l×l=7×7 and  n=8. Figure 3 provides some examples of facial images and their respective BSIF representations (with l×l=7×7 and  n=8).

Like LBP and LPQ methodologies, the BSIF codes’ co-occurrences are collected in a histogram H1, which is employed as a feature vector.

However, the simple BSIF operator based on a single block does not possess information that dominates the texture characteristics, which is forceful for the image’s occlusion and rotation. To address those limitations, an extension of the basic BSIF, the Multi-Block BSIF (MB-BSIF), is used. The concept is based on partitioning the original image into non-overlapping blocks. An undefined facial image may be split equally along the horizontal and vertical directions. As an illustration, we can derive 1, 4, or 16 blocks by segmenting the image into grids of 1 × 1, 2 × 2, or 4 × 4, as shown in Figure 4. Each block possesses details about its composition, such as the nose, eyes, or eyebrows. Overall, these blocks provide information about position relationships, such as nose to mouth and eye to eye. The blocks and the data between them are thus essential for SSFR tasks. 

Our idea was to segment the image into equal non-overlapping blocks and calculate the BSIF operator’s histograms related to the different blocks. The histogram H2 represents the fusion of the regular histograms calculated for the different blocks, as shown in Figure 5.

In the face recognition literature, some works have concentrated solely on analyzing the luminance details of facial images (i.e., grayscale). This paper suggests a different and exciting technique that exploits color texture information and shows that analysis of chrominance can be beneficial to SSFR systems. To prove this idea, we can separate the RGB facial image into three channels (i.e., red, green, and blue) and then compute the MB-BSIF separately for each channel. The final feature vector is the concatenation of their histograms in a global histogram  H3. This approach is called Multi-Block Color BSIF (MB-C-BSIF). Figure 5 provides a schematic illustration of the proposed MB-C-BSIF framework.

We note that the RGB is the most commonly employed color-space for detecting, modeling, and displaying color images. Nevertheless, its use in image interpretation is restricted due to the broad connection between the three color channels (i.e., red, green, and blue) and the inadequate separation of details in terms of luminance and chrominance. To identify captured objects, the various color channels can be highly discriminative and offer excellent contrast for several visual indicators from natural skin tones. In addition to the RGB, we studied and tested two additional color-spaces—HSV and YCbCr—to exploit color texture details. These color-spaces are based on separating components of the chrominance and luminance. For the HSV color-space, the dimensions of hue and saturation determine the image’s chrominance while the dimension of brightness (v) matches the luminance. The YCbCr color-space divides the components of the RGB into luminance (Y), chrominance blue (Cb), and chrominance red (Cr). We should note that the representation of chrominance components in the HSV and YCbCr domains is dissimilar, and consequently, they can offer additional color texture descriptions for SSFR systems.

### 3.3. K-Nearest Neighbors (K-NN) Classifier

During the classification process, each tested facial image is compared with those saved in the dataset. To assign the corresponding label (i.e., identity) to the tested image, we used the K-NN classifier associated with a distance metric. In scenarios of general usage, K-NNs show excellent flexibility and usability in substantial applications.

Technically speaking, for a presented training set $\;\left\{\left(x_{i},y_{i}\right)│\;i=1,2,\ldots ,s\right\}$, where $\;x_{i}\in \;R^{D}$ denotes the $\;i^{th}$ person’s feature vector,$\;y_{i}$ denotes this person’s label,  D is the dimension of the characteristic vector, and  s represents the number of persons. For a test person $\;x^{\prime }\in R^{D}$ that is expected to be classified, the K-NN is used to determine a training person $\;x^{*}\;$ resembling to $x^{\prime }$ based on the distance rate and then attribute the label of $\;x^{*}\;$ to $\;x^{\prime }$.

K-NN can be implemented with various distance measurements. We evaluated and compared three widely used distance metrics in this work: Hamming, Euclidean, and city block (also called Manhattan distance).

The Hamming distance between $\;x^{\prime }$ and $\;x_{i}\;$ is calculated as follows:(3)$$d\left(x^{\prime },x_{i}\right)\;=\;\overset{D}{\underset{j=1}{\sum }}│{\left(x_{j}^{\prime }-x_{ij}\right)}^{2}│$$

The Euclidean distance between $\;x^{\prime }$ and $\;x_{i}\;$ is formulated as follows:(4)$$d\left(x^{\prime },x_{i}\right)=\sqrt{\sum_{j=1}^{D}{\left(x_{j}^{\prime }-x_{ij}\right)}^{2}}$$

The city block distance between $\;x^{\prime }$ and $\;x_{i\;}$ is measured as follows: (5)$$d\left(x^{\prime },x_{i}\right)=\;\overset{D}{\underset{j=1}{\sum }}│\left(x_{j}^{\prime }-x_{ij}\right)│$$
where $x^{\prime }$ and $x_{i}$ are two vectors of dimension D, while $x_{ij}\;$ is the jth feature of $x_{i}$, and $x_{j}^{\prime }$ is the jth feature of $x^{\prime }$.

The corresponding label of $x^{\prime }$ can be determined by:(6)$$y^{\prime }=y_{i^{*}}$$
where
(7)$$i^{*}=arg_{i=1,\ldots ,s}\left(min\left(d\left(x^{\prime },x_{i}\right)\right)\right)$$

The distance metric in SSFR corresponds to calculating the similarities between the test example and the training examples.

The Algorithm 1 sums up our proposed method of SSFR recognition.

**Algorithm 1** SSFR based on MB-C-BSIF and K-NN **Input:** Facial image *X*1.   Apply histogram normalization on *X*2.   Apply median filtering on *X*3.   Divide *X* into three components (red, green, blue): Cn;n=1,2,34.   **for**
n=1 to 35.   Divide Cn into K equivalent blocks: Ckn;k=1,….,K6.   **for**
k=1 to K7.   Compute BSIF on the block-component Ckn: H1(k)(n)8.   **end for**9.   Concatenate the computed MB-BSIF features of the component Cn:10.   H2(n)=H1(1)(n)+H1(2)(n)+⋯+H1(K)(n)11.   **end for**12.   Concatenate the computed MB-C-BSIF features: H3=H2(1)+H2(2)+H2(3)13.   Apply K-NN associated with a metric distance**Output:** Identification decision

## 4. Experimental Analysis

The proposed SSFR was evaluated using the unconstrained Alex and Robert (AR) [51] and Labeled Faces in the Wild (LFW) [52] databases. In this section, we present the specifications of each utilized database and their experimental setups. Furthermore, we analyze the findings obtained from our proposed SSFR method and compare the accuracy of recognition with other current state-of-the-art approaches.

### 4.1. Experiments on the AR Database

#### 4.1.1. Database Description

The Alex and Robert (AR) face database [51] includes more than 4000 colored facial photographs of 126 individuals (56 females and 70 males); each individual has 26 different images with a frontal face taken with several facial expressions, lighting conditions, and occlusions. These photographs were acquired at an interval of two-weeks and their analysis was in two sessions (shots 1 and 2). Each session comprised 13 facial photographs per subject. A subset of facial photographs of 100 distinct individuals (50 males and 50 females) was selected in the subsequent experiments. Figure 6 displays the 26 facial images of the first individual from the AR database, along with detailed descriptions of them.

#### 4.1.2. Setups

To determine the efficiency of the proposed MB-C-BSIF in dealing with changes in facial expression, subset A (normal-1) was used as the training set and subsets B (smiling-1), C (angry-1), D (screaming-1), N (normal-2), O (smiling-2), P (angry-2), and Q (screaming-2) were employed for the test set. The facial images from the eight subsets displayed different facial expressions and were used in two different sessions. For the training set, we employed 100 images of the normal-1 type (100 images for 100 persons, i.e., one image per person). Moreover, we employed 700 images in the test set (smiling-1, angry-1, screaming-1, normal-2, smiling-2, angry-2, and screaming-2). These 700 images were divided into seven subsets for testing, with each subset containing 100 images.

As shown in Figure 6, two forms of occlusion are found in 12 subsets. The first is occlusion by sunglasses, as seen in subsets H, I, J, U, V, and W, while the second is occlusion by a scarf in subsets K, L, M, X, Y, and Z. In these 12 subsets, each individual’s photographs have various illumination conditions and were acquired in two distinct stages. There are 100 different items in each subset and the total number of facial photographs used in the test set was 1200. To examine the performance of the suggested MB-C-BSIF under conditions of object occlusion, we considered subset A as the training set and the 12 occlusion subjects as the test set, which was similar to the initial setup.

#### 4.1.3. Experiment #1 (Effects of BSIF Parameters)

As stated in Section 3.2, the BSIF operator is based on two parameters: filter kernel size l×l and bit string length  n. In this test, we assessed the proposed method by testing various BSIF parameters to obtain the best configuration, i.e., the one that yielded the best recognition accuracy. We transformed the image into a grayscale level, we did not segment the image into non-overlapping blocks (i.e., 1 × 1 block), and we used the city block distance associated with K-NN. Table 1, Table 2 and Table 3 show comprehensive details and comparisons of results obtained using some (key) BSIF configurations for facial expression variation subsets, occlusion subsets for sunglasses, and occlusion subsets for scarfs, respectively. The best results are in bold.

We note that using the parameters l×l=17×17  and n=12 for the BSIF operator achieves the best performance in identification compared to other configurations considered in this experiment. Furthermore, an increase in the identification rate appears when we augment the values of l or n. The implemented configuration can achieve better accuracy for changes in facial expression with all seven subsets. However, for subset Q, which is characterized by considerable variation in facial expression, the accuracy of recognition was very low (71%). Lastly, the performance of this implemented configuration under conditions of occlusion by an object is unsatisfactory, especially with occlusion by a scarf, and needs further improvement.

#### 4.1.4. Experiment #2 (Effects of Distance)

In this experiment, we evaluated the last configuration (i.e., grayscale level image, 1 × 1 block  l×l=17×17, and n=12) by checking various distances associated with K-NN for classification. Table 4, Table 5 and Table 6 compare the results achieved by adopting the city block distance and other well-known distances with facial expression variation subsets, occlusion subsets for sunglasses, and occlusion subsets for scarfs, respectively. The best results are in bold.

We note that the city block distance produced the most reliable recognition performance compared to the other distances analyzed in this test, such as the Hamming and Euclidean distances. As such, we can say that the city block distance is the most suitable for our method.

#### 4.1.5. Experiment #3 (Effects of Image Segmentation)

To improve recognition accuracy, especially under conditions of occlusion, we proposed decomposing the image into several non-overlapping blocks, as discussed in Section 3.2. The objective of this test was to estimate identification performance when MB-BSIF features are used instead of their global computation over an entire image. In this paper, three methods for image segmentation are considered and compared. Each original image was divided into 1 × 1 (i.e., global information), 2 × 2, and 4 × 4 blocks (i.e., local information). In other terms, an image was divided into 1 block (i.e., the original image), 4 blocks, and 16 blocks. For the last two cases, the feature vectors (i.e., histograms H1) derived from each block were fused to create the entire image extracted feature vector (Histogram H2). Table 7, Table 8 and Table 9 present and compare the recognition accuracy of the tested MB-BSIF for various blocks with subsets of facial expression variation, occlusion subsets for sunglasses, and occlusion subsets for scarfs, respectively (with grayscale images, city block distance, l×l=17×17, and n=12). The best results are in bold.

From the resulting outputs, we can observe that:-For subsets of facial expression variation, a small change arises because the results of the previous experiment were already reasonable (e.g., subsets A, B, D, N, and P). However, the accuracy rises from 71% to 76% for subset Q, which is characterized by significant changes in facial expression.-For occluded subsets, there was a significant increase in recognition accuracy when the number of blocks was augmented. As an illustration, when we applied 1 to 16 patches, the accuracy grew from 31% to 71% for subset Z, from 46% to 79% for subset W, and from 48% to 84% for subset Y.-As such, in the case of partial occlusion, we may claim that local information is essential. It helps to go deeper in extracting relevant information from the face like details about the facial structure, such as the nose, eyes, or mouth, and information about position relationships, such as nose to mouth, eye to eye, and so on.-Finally, we note that the 4 × 4 blocks provided the optimum configuration with the best accuracy for subsets of facial expression, occlusion by sunglasses, and scarf occlusion.

#### 4.1.6. Experiment #4 (Effects of Color Texture Information)

For this analysis, we evaluated the performance of the last configuration (i.e., segmentation of the image into 4 × 4 blocks, K-NN associated with city block distance, l×l=17×17, and n=12) by testing three color-spaces, namely, RGB, HSV, and YCbCr, instead of transforming the image into grayscale. This feature extraction method is called MB-C-BSIF, as described in Section 3.2. The AR database images are already in RGB and so do not need a transformation of the first color-space. However, the images must be converted from RGB to HSV and RGB to YCbCr for the other color-spaces. Table 10, Table 11 and Table 12 display and compare the recognition accuracy of the MB-C-BSIF using several color-spaces with subsets of facial expression variations, occlusion by sunglasses, and occlusion by a scarf, respectively. The best results are in bold.

From the resulting outputs, we can see that:-The results are almost identical for subsets of facial expression variation with all checked color-spaces. In fact, with the HSV color-space, a slight improvement is reported, although slight degradations are observed with both RGB and YCbCr color-spaces.-All color-spaces see enhanced recognition accuracy compared to the grayscale standard for sunglasses occlusion subsets. RGB is the color-space with the highest output, seeing an increase from 91.83% to 93.50% in terms of average accuracy.-HSV shows some regression for scarf occlusion subsets, but both the RGB and YCbCr color-spaces display some progress compared to the grayscale norm. Additionally, RGB remains the color-space with the highest output.-The most significant observation is that the RGB color-space saw significantly improved performance in the V, W, Y, and Z subsets (from 81% to 85% with V; 79% to 84% with W; 84% to 88% with Y; and 77% to 87% with Z). Note that images of these occluded subsets are characterized by light degradation (either to the right or left, as shown in Figure 6).-Finally, we note that the optimum color-space, providing a perfect balance between lighting restoration and improvement in identification, was the RGB.

#### 4.1.7. Comparison #1 (Protocol I)

To confirm that our suggested method produces superior recognition performance with variations in facial expression, we compared the collected results with several state-of-the-art methods recently employed to tackle the SSFR issue. Table 13 presents the highest accuracies obtained using the same subsets and the same assessment protocol with Subset A as the training set and subsets of facial expression variations B, C, D, N, O, and P constituting the test set. The results presented in Table 13 are taken from several references [36,39,53,54]. “- -” signifies that the considered method has no experimental results. The best results are in bold.

The outcomes obtained validate the robustness and reliability of our proposed SSFR system compared to state-of-the-art methods when assessed with identical subsets. We suggest a competitive technique that has achieved a desirable level of identification accuracy with the six subsets of up to: 100.00% for B and C; 95.00% for D; 97.00% for N; 92.00% for O; and 93.00% for P.

For all subsets, our suggested technique surpasses the state-of-the-art methods analyzed in this paper, i.e., the proposed MB-C-BSIF can achieve excellent identification performance under the condition of variation in facial expression.

#### 4.1.8. Comparison #2 (Protocol II)

To further demonstrate the efficacy of our proposed SSFR system, we also compared the best configuration of the MB-C-BSIF (i.e., RGB color-space, segmentation of the image into 4 × 4 blocks, city block distance, l×l=17×17, and n=12) with recently published work under unconstrained conditions. We followed the same experimental protocol described in [33,39]. Table 14 displays the accuracies of the works compared on the tested subsets H + K (i.e., occlusion by sunglasses and scarf) and subsets J + M (i.e., occlusion by sunglasses and scarf with variations in lighting). The best results are in bold.

In Table 14, we can observe that the work presented by Zhu et al. [33], called LGR, shows a comparable level, but the identification accuracy of our MB-C-BSIF procedure is much higher than all the methods considered for both test sessions.

Compared to related SSFRs, which can be categorized as either generic learning methods (e.g., ESRC [31], SVDL [32], and LGR [33], image partitioning methods (e.g., CRC [35], PCRC [34], and DNNC [39]) or deep learning methods (e.g., DCNN [41] and BDL [44]), the capabilities of our method can be explained in terms of its exploitation of different forms of information. This can be summarized as follows:-The BSIF descriptor scans the image pixel by pixel, i.e., we consider the benefits of local information.-The image is decomposed into several blocks, i.e., we exploit regional information.-BSIF descriptor occurrences are accumulated in a global histogram, i.e., we manipulate global information.-The MB-BSIF is applied to all RGB image components, i.e., color texture information is exploited.

**Table 14 sensors-21-00728-t014:** Comparison of 12 methods on occlusion and lighting-occlusion sessions.

Authors	Year	Method	Occlusion (H + K) (%)	Lighting + Occlusion (J + M) (%)	Average Accuracy (%)
Zhang et al. [35]	2011	CRC	58.10	23.80	40.95
Deng et al. [31]	2012	ESRC	83.10	68.60	75.85
Zhu et al. [34]	2012	PCRC	95.60	81.30	88.45
Yang et al. [32]	2013	SVDL	86.30	79.40	82.85
Lu et al. [36]	2012	DMMA	46.90	30.90	38.90
Zhu et al. [33]	2014	LGR	98.80	96.30	97.55
Ref. [67]	2016	SeetaFace	63.13	55.63	59.39
Zeng et al. [41]	2017	DCNN	96.5	88.3	92.20
Chu et al. [65]	2019	MFSA+	91.3	79.00	85.20
Cuculo et al. [68]	2019	SSLD	90.18	82.02	86.10
Zhang et al. [39]	2020	DNNC	92.50	79.50	86.00
Du and Da [44]	2020	BDL	93.03	91.55	92.29
**Our method**	**2021**	**MB-C-BSIF**	**99.5**	**98.5**	**99.00**

To summarize this first experiment, the performance of the proposed approach was evaluated using the AR database. In this experiment, the issues studied were changes in facial expression, lighting and occlusion by sunglasses and headscarf, which are the most common cases in real-world applications. As presented in Table 13 and Table 14, our system obtained very good results (i.e., 96.17% with Protocol I and 99% with Protocol II) that surpass all the approaches compared (including the handcrafted and deep-learning-based approaches), i.e., that the approach we propose is appropriate and effective in the presence of the problems mentioned above.

### 4.2. Experiments on the LFW Database

#### 4.2.1. Database Description

The Labeled Faces in the Wild (LFW) database [52] comprises more than 13,000 photos collected from the World Wide Web of 5749 diverse subjects in challenging situations, of which 1680 subjects possess two or more shots per individual. Our tests employed the LFW-a, a variant of the standard LFW where the facial images are aligned with a commercial normalization tool. It can be observed that the intra-class differences in this database are very high compared to the well-known constrained databases and face normalization has been carried out. The size of each image is 250 × 250 pixels and uses the jpeg extension. LFW is a very challenging database: it aims to investigate the unconstrained issues of face recognition, such as changes in lighting, age, clothing, focus, facial expression, color saturation, posture, race, hairstyle, background, camera quality, gender, ethnicity, and other factors, as presented in Figure 7.

#### 4.2.2. Experimental Protocol

This study followed the experimental protocol presented in [30,32,33,34]. From the LFW-a database, we selected only those subjects possessing more than 10 images to obtain a subset containing the facial images of 158 individuals. We cropped each image to a size of 120 × 120 pixels and then resized it to 80 × 80 pixels. We considered the first 50 subjects’ facial photographs to create the training set and the test set. We randomly selected one shot from each subject for the training set, while the remaining images were employed in the test set. This process was repeated for five permutations and the average result for each was taken into consideration.

#### 4.2.3. Limitations of SSFR Systems

In this section, the SSFR systems, and particularly the method we propose, will be voluntarily tested in a situation that is not adapted to their application: they are applicable in the case where only one sample is available and, very often, this sample is captured in very poor conditions.

We are particularly interested in cases where hundreds of samples are available, as in the LFW database, or when the training stage is based on millions of samples. In such a situation, deep learning approaches must be obviously chosen.

Therefore, the objective of this section is to assess the limitations of our approach.

Table 15 summarizes the performance of several rival approaches in terms of identification accuracy. Our best result was obtained by adopting the following configuration:
-BSIF descriptor with filter size l×l=17×17 and bit string length n=12.-K-NN classifier associated with city block distance.-Segmentation of the image into blocks of 40 × 40 and 20 × 20 pixels.-RGB color-space.

We can observe that the traditional approaches did not achieve particularly good identification accuracies. This is primarily because the photographs in the LFW database have been taken in unregulated conditions, which generates facial images with rich intra-class differences and increases face recognition complexity. As a consequence, the efficiency of the SSFR procedure is reduced. However, our recommended solution is better than the other competing traditional approaches. The superiority of our method can be explained by its exploitation of different forms of information, namely: local, regional, global, and color texture information. SVDL [32] and LGR [33] also achieved success in SSFR because the intra-class variance information obtained from other subjects in the standardized training set (i.e., augmenting the training-data) helped boost the performance of the system. Additionally, KNNMMDL [30] achieved good performance because it uses the Weber-face algorithm in the preprocessing step, which handles the illumination variation issue and employs data augmentation to enrich the intra-class variation in the training set.

In another experiment, we implemented and tested the successful DeepFace algorithm [12], whose weights were trained on millions of images from the ImageNet database that are close to real-life situations. As presented in Table 15, the DeepFace algorithm shows significant superiority to the compared methods. This success is down to the profound and specific training of the weights in addition to the significant number of images employed in its operation.

In a recent work by Zeng et al. [72], the authors combined traditional (handcrafted) and deep learning (TDL) characteristics to overcome the limitation of each class. They reached an identification accuracy of near 74%, which is something of a quantum leap in this challenging topic.

In the comparative study presented in [73], we can see that current face recognition systems employing several examples in the training set achieve very high accuracy with the LFW database, especially with deep-learning-based methods. However, SSFR systems suffer considerably when using the challenging LFW database and further research is required to improve their reliability.

In the situation where the learning stage is based on millions of images, the proposed SSFR technique cannot be used. In such a situation, References [12,72], which use deep learning techniques with data augmentation [12] or deep learning features combined with handcrafted features [72], allow one to obtain better accuracy.

Finally, the proposed SSFR method is reserved for the case where only one sample per person is available, which is the most common case in the real world through remote surveillance or unmanned aerial vehicles’ shots. In these applications, faces are most often captured under harsh conditions, such as changing lighting, posture, or if the person is wearing accessories such as glasses, masks, or disguises. In these cases, the method proposed here is by far the most accurate. Finally, it would be interesting to explore and test some proven approaches that have shown good performance in solving real-world problems, in order to evaluate their performance using the same protocol and database, such as multi-scale principal component analysis (MSPCA) [74], signal decomposition methods [75,76], generative adversarial neural networks (GAN) [77], and centroid-displacement-based-K-NN [78].

## 5. Conclusions and Perspectives

In this paper, we have presented an original method for Single-Sample Face Recognition (SSFR) based on the Multi-Block Color-binarized Statistical Image Features (MB-C-BSIF) descriptor. It allows for the extraction of features for classification by the K-nearest neighbors (K-NN) method. The proposed method exploits various kinds of information, including local, regional, global, and color texture information. In our experiments, the MB-C-BSIF has been evaluated on several subsets of images from the unconstrained AR and LFW databases. Experiments conducted on the AR database have shown that our method significantly improves the performance of SSFR classification when dealing with several variations of facial recognition. The proposed feature extraction strategy achieves a high accuracy, with an average value of 96.17% and 99% for the AR database with Protocols I and II, respectively. These significant results validate the effectiveness of the proposed method compared to state-of-the-art methods. The potential applications of the method are oriented towards a computer-aided technology that can be used for real-time identification.

In the future, we aim to explore the effectiveness of combining both deep learning and traditional methods in addressing the SSFR issue. Hybrid features combine handcrafted features with deep characteristics to collect richer information than those obtained by a single feature extraction method, thus improving the level of recognition. Besides, we plan to develop a deep learning method based on semantic information, such as age, gender, and ethnicity, to solve the problem of SSFR, which is an area that deserves further study. We also aim to investigate and analyze the SSFR issue in unconstrained environments using large-scale databases that hold millions of facial images.

## Figures and Tables

**Figure 1 sensors-21-00728-f001:**
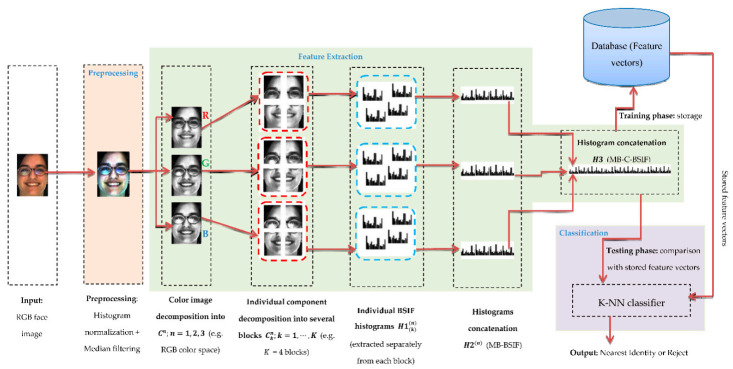
Schematic of the proposed Single-Sample Face Recognition (SSFR) system based on the Multi-Block Color-Binarized Statistical Image Features (MB-C-BSIF) descriptor.

**Figure 2 sensors-21-00728-f002:**
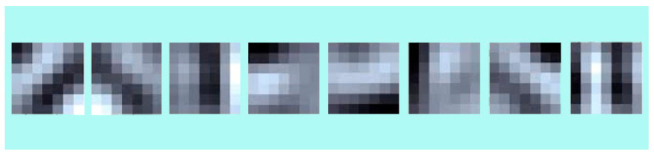
Examples of 7 × 7 BSIF filter banks learned from natural pictures.

**Figure 3 sensors-21-00728-f003:**
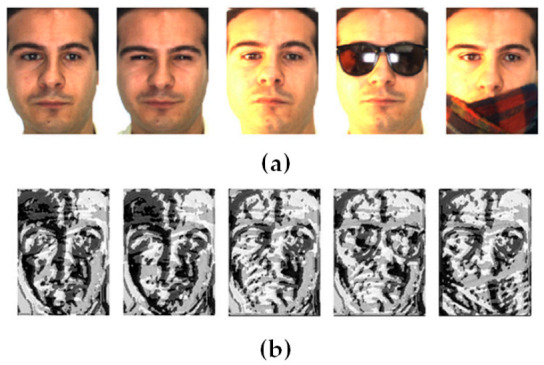
(**a**) Examples of facial images, and (**b**) their parallel BSIF representations.

**Figure 4 sensors-21-00728-f004:**
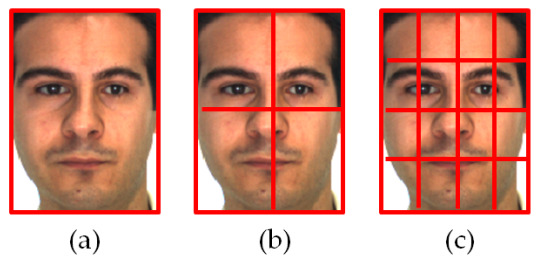
Examples of multi-block (MB) image decomposition: (**a**) 1 × 1, (**b**) 2 × 2, and (**c**) 4 × 4.

**Figure 5 sensors-21-00728-f005:**
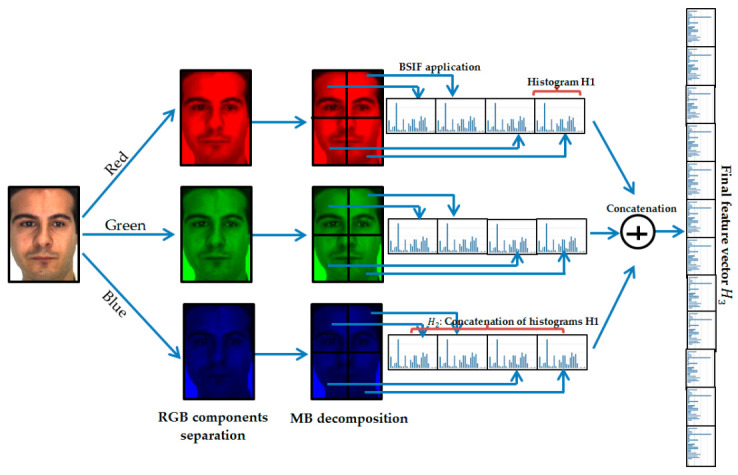
Structure of the proposed feature extraction approach: MB-C-BSIF.

**Figure 6 sensors-21-00728-f006:**
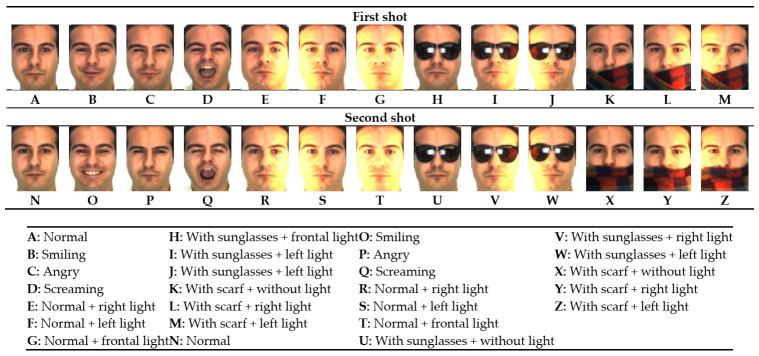
The 26 facial images of the first individual from the AR database and their detailed descriptions.

**Figure 7 sensors-21-00728-f007:**
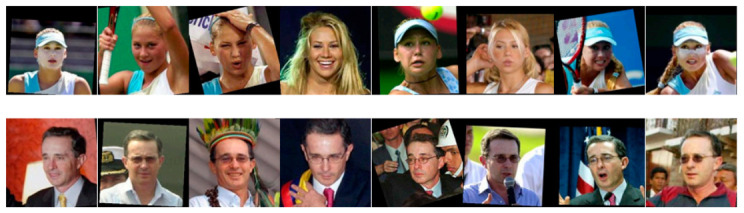
Examples of two different subjects from the Labeled Faces in the Wild (LFW)-a database.

**Table 1 sensors-21-00728-t001:** Comparison of the results obtained using six BSIF configurations with changes in facial expression.

l×l (Pixels)	$n\;\left(\mathrm{Bits}\right)$	Accuracy (%)							Average Accuracy (%)
		B	C	D	N	O	P	Q	
3 × 3	5	70	72	38	36	20	24	14	39.14
5 × 5	9	94	97	59	75	60	66	30	68.71
9 × 9	12	100	100	91	95	90	92	53	88.71
11 × 11	8	97	99	74	85	70	75	43	77.57
15 × 15	12	100	100	96	97	96	96	73	94.00
17 × 17	12	100	100	98	97	96	97	71	94.14

**Table 2 sensors-21-00728-t002:** Comparison of the results obtained using six BSIF configurations with occlusion by sunglasses.

l×l (Pixels)	$n\;\left(\mathrm{Bits}\right)$	Accuracy (%)						Average Accuracy (%)
		H	I	J	U	V	W	
3 × 3	5	29	8	4	12	4	3	10.00
5 × 5	9	70	24	14	28	14	8	26.50
9 × 9	12	98	80	61	80	38	30	61.50
11 × 11	8	78	34	23	48	26	15	37.33
15 × 15	12	100	84	85	87	50	46	75.33
17 × 17	12	100	91	87	89	58	46	78.50

**Table 3 sensors-21-00728-t003:** Comparison of the results obtained using six BSIF configurations with occlusion by scarf.

l×l (Pixels)	$n\;\left(\mathrm{Bits}\right)$	Accuracy (%)						Average Accuracy (%)
		K	L	M	X	Y	Z	
3 × 3	5	7	4	2	3	2	2	3.33
5 × 5	9	22	9	6	12	6	2	9.50
9 × 9	12	88	54	34	52	31	15	45.67
11 × 11	8	52	12	90	22	9	7	32.00
15 × 15	12	97	69	64	79	48	37	65.67
17 × 17	12	98	80	63	90	48	31	68.33

**Table 4 sensors-21-00728-t004:** Comparison of the results obtained using different distances with changes in facial expression.

Distance	Accuracy (%)							Average Accuracy (%)
	B	C	D	N	O	P	Q	
Hamming	63	79	9	69	23	40	6	41.29
Euclidean	99	100	80	90	83	82	43	82.43
City block	100	100	98	97	96	97	71	94.14

**Table 5 sensors-21-00728-t005:** Comparison of the results obtained using different distances with occlusion by sunglasses.

Distance	Accuracy (%)						Average Accuracy (%)
	H	I	J	U	V	W	
Hamming	37	5	6	11	4	2	10.83
Euclidean	96	68	42	68	31	17	53.67
City block	100	91	87	89	58	46	78.50

**Table 6 sensors-21-00728-t006:** Comparison of the results obtained using different distances with occlusion by scarf.

Distance	Accuracy (%)						Average Accuracy (%)
	K	L	M	X	Y	Z	
Hamming	34	5	8	20	4	4	12.50
Euclidean	79	32	16	41	22	5	32.50
City block	98	80	63	90	48	31	68.33

**Table 7 sensors-21-00728-t007:** Comparison of the results obtained using different divided blocks with changes in facial expression.

Segmentation	Accuracy (%)							Average Accuracy (%)
	B	C	D	N	O	P	Q	
(1 × 1)	100	100	98	97	96	97	71	94.14
(2 × 2)	100	100	95	98	92	91	60	90.86
(4 × 4)	100	100	99	98	92	97	76	94.57

**Table 8 sensors-21-00728-t008:** Comparison of the results obtained using different divided blocks with occlusion by sunglasses.

Segmentation	Accuracy (%)						Average Accuracy (%)
	H	I	J	U	V	W	
(1 × 1)	100	91	87	89	58	46	78.50
(2 × 2)	100	99	98	91	83	71	90.33
(4 × 4)	100	99	99	93	81	79	91.83

**Table 9 sensors-21-00728-t009:** Comparison of the results obtained using different divided blocks with occlusion by scarf.

Segmentation	Accuracy (%)						Average Accuracy (%)
	K	L	M	X	Y	Z	
(1 × 1)	98	80	63	90	48	31	68.33
(2 × 2)	98	95	92	92	79	72	88.00
(4 × 4)	99	98	95	93	84	77	91.00

**Table 10 sensors-21-00728-t010:** Comparison of the results obtained using different color-spaces with changes in facial expression.

Color-Space	Accuracy (%)							Average Accuracy (%)
	B	C	D	N	O	P	Q	
Gray Scale	100	100	99	98	92	97	76	94.57
RGB	100	100	95	97	92	93	67	92.00
HSV	100	100	99	97	96	95	77	94.86
YCbCr	100	100	96	98	93	93	73	93.29

**Table 11 sensors-21-00728-t011:** Comparison of the results obtained using different color-spaces with occlusion by sunglasses.

Color-Space	Accuracy (%)						Average Accuracy (%)
	H	I	J	U	V	W	
Gray Scale	100	99	99	93	81	79	91.83
RGB	100	99	100	93	85	84	93.50
HSV	100	97	99	96	82	80	92.33
YCbCr	100	99	98	93	81	80	91.83

**Table 12 sensors-21-00728-t012:** Comparison of the results obtained using different color-spaces with occlusion by scarf.

Color-Space	Accuracy (%)						Average Accuracy (%)
	K	L	M	X	Y	Z	
Gray Scale	99	98	95	93	84	77	91.00
RGB	99	97	97	94	88	81	92.67
HSV	99	96	90	95	75	74	88.17
YCbCr	98	98	96	93	87	78	91.67

**Table 13 sensors-21-00728-t013:** Comparison of 18 methods of facial expression variation subsets.

Authors	Year	Method	Accuracy						Average Accuracy (%)
			B	C	D	N	O	P	
Turk, Pentland [55]	1991	PCA	97.00	87.00	60.00	77.00	76.00	67.00	77.33
Wu and Zhou [56]	2002	(PC)^2^A	97.00	87.00	62.00	77.00	74.00	67.00	77.33
Chen et al. [57]	2004	E(PC)^2^A	97.00	87.00	63.00	77.00	75.00	68.00	77.83
Yang et al. [58]	2004	2DPCA	97.00	87.00	60.00	76.00	76.00	67.00	77.17
Gottumukkal and Asari [59]	2004	Block-PCA	97.00	87.00	60.00	77.00	76.00	67.00	77.33
Chen et al. [60]	2004	Block-LDA	85.00	79.00	29.00	73.00	59.00	59.00	64.00
Zhang and Zhou [61]	2005	(2D)^2^PCA	98.00	89.00	60.00	71.00	76.00	66.00	76.70
Tan et al. [62]	2005	SOM	98.00	88.00	64.00	73.00	77.00	70.00	78.30
He et al. [63]	2005	LPP	94.00	87.00	36.00	86.00	74.00	78.00	75.83
Zhang et al. [27]	2005	SVD-LDA	73.00	75.00	29.00	75.00	56.00	58.00	61.00
Deng et al. [64]	2010	UP	98.00	88.00	59.00	77.00	74.00	66.00	77.00
Lu et al. [36]	2012	DMMA	99.00	93.00	69.00	88.00	85.00	85.50	79.00
Mehrasa et al. [53]	2017	SLPMM	99.00	94.00	65.00	- -	- -	- -	- -
Ji et al. [54]	2017	CPL	92.22	88.06	83.61	83.59	77.95	72.82	83.04
Zhang et al. [37]	2018	DMF	100.00	99.00	66.00	- -	- -	- -	- -
Chu et al. [65]	2019	MFSA+	100.00	100.00	74.00	93.00	85.00	86.00	89.66
Pang et al. [66]	2019	RHDA	97.08	97.00	96.25	- -	- -	- -	- -
Zhang et al. [39]	2020	DNNC	100.00	98.00	69.00	92.00	76.00	85.00	86.67
**Our method**	**2021**	**MB-C-BSIF**	**100.00**	**100.00**	**95.00**	**97.00**	**92.00**	**93.00**	**96.17**

**Table 15 sensors-21-00728-t015:** Identification accuracies using the LFW database.

Authors	Year	Method	Accuracy (%)
Chen et al. [60]	2004	Block LDA	16.40
Zhang et al. [27]	2005	SVD-FLDA	15.50
Wright et al. [69]	2009	SRC	20.40
Su et al. [70]	2010	AGL	19.20
Zhang et al. [35]	2011	CRC	19.80
Deng et al. [31]	2012	ESRC	27.30
Zhu et al. [34]	2012	PCRC	24.20
Yang et al. [32]	2013	SVDL	28.60
Lu et al. [36]	2012	DMMA	17.80
Zhu et al. [33]	2014	LGR	30.40
Ji et al. [54]	2017	CPL	25.20
Dong et al. [30]	2018	KNNMMDL	32.30
Chu et al. [65]	2019	MFSA+	26.23
Pang et al. [66]	2019	RHDA	32.89
Zhou et al. [71]	2019	DpLSA	37.55
Our method	2021	MB-C-BSIF	38.01
Parkhi et al. [12]	2015	Deep-Face	62.63
Zeng et al. [72]	2018	TDL	74.00

## Data Availability

Data sharing not applicable.

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
