# Peer review of "Multi-Block Color-Binarized Statistical Images for Single-Sample Face Recognition"

_sensors, 2021, doi:10.3390/s21030728_

Round 1

Reviewer 1 Report

1.  Please outline some of the practical applications and limitations of the work2.  Do the authors have any plans to make the code and data-set open-source?3.  What is the statistical power of the study?

  1. What are the significant advantage of the proposed method against previously published ones?
  2. Why not authors used multiscale principal component analysis (MSPCA) which is a combination of PCA and wavelets and useful for image analysis denoising? Authors should discuss this method in introduction section. The details of MSPCA can be found in “Motor imagery BCI classification based on novel two-dimensional modelling in empirical wavelet transform
  3. Why authors not used transfer learning based techniques for classification purpose? Or atleast discuss latest techniques which can be found in Exploiting Multiple Optimizers with Transfer Learning Techniques for the Identification of COVID-19 Patients
  4. The authors did not include signal decomposition methods in the introduction part which plays key role in image classification. I recommend authors to have a look on following article “Motor Imagery EEG Signals Classification Based on Mode Amplitude and Frequency Components Using Empirical Wavelet Transform
  5. The combination of signal decomposition with dimension reduction techniques along with neural networks also provide good accuracy for both subject dependent and independent motor imagery based BCI systems. They authors need to discuss this issue, detail may be found in “Exploiting dimensionality reduction and neural network techniques for the development of expert brain–computer interfaces”.
  6. Please provide the details of future direction and possible solutions to continue this topic.
  • Finally, I suggest authors to sit with English native speaker to improve the writing of proposed work.

Reviewer 2 Report

Besides using k-NN, the authors are recommended to test the proposed framework with an enhanced k-NN algorithm presented in 10.1109/THMS.2015.2453203 (DOI). An unofficial implementation of that algorithm can be found from this link: https://github.com/SonDaoDuy/Centroid-Displacement-based-k-NN

In addition, some recent one-shot learning methods should be included and compared in the manuscript.

A major revision is needed for this manuscript.

Reviewer 3 Report

The paper deals with an interesting topic face recognition method based on an image processing. In the present form the paper presents many lacks. Starting from the introduction authors should provide a wider picture on the research domain, also for instance 3D approaches rather than only 2D ones, in order to better justify the proposed works. Some more references should be added as for instance the following ones:
-VEZZETTI, Enrico, et al. 3D geometry-based automatic landmark localization in presence of facial occlusions. Multimedia Tools and Applications, 2018, 77.11: 14177-14205.
-ECHEAGARAY-PATRON, et al. Conformal parameterization and curvature analysis for 3D facial recognition. In: 2015 International Conference on Computational Science and Computational Intelligence (CSCI). IEEE, 2015. p. 843-844.

Also the methodological section should be improved by the usage of a clear graphical flowchart that could provide a clear picture on the overall method for going further with more details on the specific stages of the method. Regarding the experimental validation some more issues should be provided for what concerns the experimental setting. This is very important in order to better understand the results analysis and proposed method advantages and disadvantages

Round 2

Reviewer 1 Report

i am satisfied with the revisions.

Reviewer 2 Report

I am OK with the response from the authors. The manuscript can be considered for an acceptance.